# Weekly Cisplatin and 5-Fluorouracil in Neoadjuvant Chemoradiotherapy for Esophageal Cancer: A Pandemic-Era Evaluation

**DOI:** 10.3390/medicina61081326

**Published:** 2025-07-23

**Authors:** Yi-Ting Hwang, Cheng-Yen Chuang, Chien-Chih Chen

**Affiliations:** 1Department of Radiation Oncology, Taichung Veterans General Hospital, Taichung 407219, Taiwan; ythwang@vghtc.gov.tw; 2Division of Thoracic Surgery, Department of Surgery, Taichung Veterans General Hospital, Taichung 407219, Taiwan; chuang5045@vghtc.gov.tw; 3Department of Medical Imaging and Radiological Sciences, Central Taiwan University of Science and Technology, Taichung 406053, Taiwan

**Keywords:** weekly cisplatin and 5-fluorouracil, neoadjuvant chemoradiotherapy, esophageal cancer, COVID-19 pandemic, squamous cell carcinoma

## Abstract

*Background and Objectives*: The COVID-19 pandemic disrupted cancer care, prompting adaptations to reduce patient exposure while preserving treatment efficacy. This retrospective observational study compared a weekly cisplatin and 5-fluorouracil (5-FU) regimen to the standard monthly regimen for neoadjuvant chemoradiotherapy in patients with locally advanced esophageal squamous cell carcinoma. *Materials and Methods*: This single-center retrospective study included 91 patients, divided into two cohorts: weekly chemotherapy (n = 30) and standard chemotherapy (n = 61). Treatment assignment was based on hospital policy changes during the pandemic, with weekly outpatient chemotherapy implemented after November 2022 to conserve inpatient resources. All patients received radiotherapy at 50 Gy in 25 fractions. The weekly regimen consisted of cisplatin 20 mg/m^2^ and 5-FU 800 mg/m^2^, administered over 1–2 h weekly, while the standard regimen administered the same doses over four consecutive days on weeks 1 and 5. Primary endpoints were pathologic complete response (pCR), progression-free survival (PFS), and overall survival (OS). *Results*: The response rates were similar between groups (weekly: 86.7% vs. standard: 90.2%; *p* = 0.724). The weekly regimen group showed a higher pCR (40.0% vs. 26.2%; *p* = 0.181) and significantly lower recurrence (26.7% vs. 52.5%; *p* = 0.020). Mortality was also reduced in the weekly group (6.7% vs. 34.4%; *p* = 0.004), though the follow-up duration was shorter (10.6 vs. 22.8 months; *p* < 0.001). *Conclusions*: In this retrospective observational study, weekly cisplatin and 5-FU demonstrated comparable efficacy to the standard regimen, with potential advantages in reducing recurrence and mortality. This modified approach may be a viable alternative for maintaining oncologic outcomes while minimizing the burden on healthcare systems during pandemic conditions, although prospective validation is needed.

## 1. Introduction

Esophageal cancer is the 11th most common cancer worldwide and the 7th leading cause of cancer-related mortality [1,2]. The ChemoRadiotherapy for Oesophageal cancer Followed by Surgery (CROSS) trial has established neoadjuvant concurrent chemoradiotherapy (CCRT) followed by esophagectomy as the standard treatment for potentially operable, locally advanced esophageal cancer, demonstrating superior survival outcomes compared to esophagectomy alone [3]. Currently, two neoadjuvant CCRT regimens are in use: weekly paclitaxel and carboplatin with concurrent radiotherapy (41.4 Gy in 23 fractions) and cisplatin with fluorouracil (5-FU) alongside radiotherapy (50.4 Gy in 28 fractions). The CALGB 9781 trial specifically outlined a regimen of cisplatin 100 mg/m^2^ and fluorouracil 1000 mg/m^2^, administered consecutively for 4 days during weeks 1 and 5 in conjunction with radiotherapy [4].

The COVID-19 pandemic, caused by severe acute respiratory syndrome coronavirus 2 (SARS-CoV-2), emerged in December 2019 and had rapidly spread worldwide by March 2020. The pandemic caused significant healthcare disruptions in cancer services, with declines of 39% in cancer screening, 23% in diagnoses, 24% in diagnostic procedures, and 28% in treatment based on data from 245 studies across 46 countries [5,6]. Evidence suggests that nosocomial transmission of SARS-CoV-2 in healthcare settings occurs through droplets, aerosols, and fecal–oral or fecal droplet routes [7]. To mitigate COVID-19 transmission and reduce the burden on medical facilities, many institutions implemented policies to limit or delay non-emergent admissions. In response to these policies and to minimize the exposure risk for cancer patients, our clinical practice shifted to administering neoadjuvant chemotherapy weekly in outpatient settings, deviating from the conventional approach, which required hospitalization.

Currently, there is a lack of research addressing outpatient adaptations of chemotherapy regimens for esophageal cancer patients who received neoadjuvant CCRT during the pandemic. Most published studies have focused on modifications in radiotherapy, such as hypofractionation or acceleration, to reduce the overall treatment time and number of hospital visits. One study from the United Kingdom showed that 56.5% of centers omitted chemotherapy from concurrent chemoradiotherapy for head and neck cancer patients to limit exposure risks [8]. Our aim is to maintain chemotherapy while minimizing hospitalization. At our institution, cisplatin and 5-fluorouracil (5-FU) is the standard neoadjuvant CCRT regimen for esophageal cancer due to Taiwan’s National Health Insurance coverage’s limitations regarding carboplatin and paclitaxel. This research evaluates the safety and efficacy of weekly cisplatin and 5-FU as outpatient neoadjuvant chemotherapy for esophageal cancer, compared to the standard inpatient regimen. We aim to determine if this modified approach could be implemented during future pandemic situations, potentially preserving the benefits of concurrent chemoradiotherapy without compromising treatment outcomes or increasing infection rates.

## 2. Materials and Methods

### 2.1. Patient Selection

This retrospective observational study included patients with histologically confirmed esophageal cancer who were treated at Taichung Veterans General Hospital between 2020 and 2023. The eligibility criteria included (1) clinical stage II–IVB according to the American Joint Committee on Cancer (AJCC) Tumor–Node–Metastasis (TNM) 8th edition classification, and (2) receipt of neoadjuvant chemoradiotherapy, followed by surgical resection. The exclusion criteria were (1) immediate surgical intervention without neoadjuvant chemoradiotherapy, (2) inoperability or refusal of surgery, and (3) neoadjuvant treatment regimens deviating from the institutional protocol based on cisplatin and 5-fluorouracil.

A total of 318 patients who were newly diagnosed esophageal cancer patients were reviewed initially, and 91 patients met the inclusion criteria and were enrolled between May 2020 and December 2023 (Figure 1). The allocation of patients to treatment groups was determined by hospital policy in response to the COVID-19 pandemic. Prior to late November 2022, patients received the standard chemotherapy regimen, requiring hospitalization. However, due to the surge in COVID-19 cases in Taiwan, the hospital policy shifted to preserve beds for COVID-19 patients requiring specialized care and isolation. Consequently, from late November 2022 onwards, new patients were assigned to the weekly chemotherapy regimen, which could be administered in outpatient clinics. This strategic adjustment aimed to maintain critical medical resource capacity while ensuring the continuity of cancer care. Comprehensive pretreatment staging was performed for all patients, encompassing medical history, physical examination, esophageal tumor biopsy, chest X-ray, chest computed tomography (CT), endoscopic ultrasound, bronchoscopy, complete blood count, liver and renal function tests, and ^18^F-fluorodeoxyglucose positron emission tomography/computed tomography (^18^F-FDG PET/CT). PET/CT scans were utilized for radiation treatment planning.

This study was approved by the Institutional Review Board of Taichung Veterans General Hospital.

### 2.2. Treatment Protocol

All 91 patients underwent CT simulation in the supine position. The following volumes were delineated on CT images: gross tumor volume (GTV), clinical target volume (CTV), planning target volume (PTV), and organs at risk (OARs). The GTV encompassed the gross esophageal tumor and enlarged lymph nodes, as identified by PET/CT and/or CT scans. The CTV for the primary tumor included the GTV plus a 1 cm radial margin and a 5 cm longitudinal margin. For upper or middle esophageal tumors, the nodal CTV covered the nodal GTV with a 5 mm margin, the mediastinum, and supraclavicular regions. For lower-third esophageal tumors, the nodal CTV included the nodal GTV with a 5 mm margin and the celiac trunk area. The PTV was defined as CTV plus a 5 mm margin to account for setup errors and organ motion.

Radiation therapy consisted of 50 Gy, delivered in 25 daily fractions of 2.0 Gy, with five fractions per week. Radiotherapy was delivered using VitalBeam or RapidArc linear accelerators (Varian Medical Systems, Palo Alto, CA, USA) with a dynamic multi-leaf linear accelerator (6 MV and/or 10 MV photon energy) and the source–axis distance (SAD) technique.

Concurrent neoadjuvant chemotherapy was administered using two different schedules. In the standard group, patients received cisplatin 20 mg/m^2^ and fluorouracil 800 mg/m^2^ in two cycles: the first cycle on days 1–4 and the second cycle on days 29–32. In the weekly group, a weekly chemotherapy regimen consisting of six cycles with cisplatin 20 mg/m^2^ and fluorouracil 800 mg/m^2^ was administered on days 1, 8, 15, 22, 29, and 36. Comprehensive pre-chemotherapy evaluations, including blood profile analysis, urinalysis, and chest X-ray, were conducted before each administration to ensure patient safety and treatment tolerance.

Chemotherapy dose adjustments were implemented based on toxicity assessments utilizing the Common Terminology Criteria for Adverse Events (CTCAE) version 4.0. Continuation of treatment at the prescribed dose was maintained for toxicities deemed by the investigator as unlikely to progress to serious or life-threatening status, and which did not necessitate therapy interruption or delay. Notably, anemia management was achieved through transfusion support without necessitating dose alterations. In instances where dose interruptions were clinically indicated, they were limited to a maximum duration of two weeks, with the final decision resting on the investigator’s clinical judgment, taking into account individual patient factors and overall treatment tolerability.

Specific dose modifications were implemented based on individual patient factors and observed toxicities. The cisplatin dosage was reduced in cases of impaired renal function, while 5-fluorouracil (5-FU) adjustments were made for patients with compromised liver function or those experiencing intolerable gastrointestinal side effects. Treatment completion criteria were defined distinctly for each chemotherapy regimen based on the institutional clinical practice. In the weekly chemotherapy group, completion of CCRT was defined as the administration of at least four cycles of chemotherapy, coupled with a cumulative radiation dose within 5% of the planned dose. For the standard chemotherapy group, completion of CCRT encompassed the full administration of two chemotherapy cycles, along with a cumulative radiation dose within 5% of the planned dose. These criteria ensured a standardized approach to assessing treatment adherence and efficacy across both regimens, while allowing for necessary adjustments to manage patient-specific toxicities and maintain optimal therapeutic outcomes. To accurately assess treatment delivery and intensity, the actually administered doses of both chemotherapy and radiotherapy were meticulously calculated and analyzed for each patient.

### 2.3. Surgical Procedure

Surgical intervention was performed 4 to 6 weeks after the completion of neoadjuvant chemoradiotherapy. The procedure consisted of thoracoscopic esophagectomy with lymph node dissection, followed by esophageal reconstruction using a gastric tube.

### 2.4. Histopathological Evaluation

Pathological examinations were conducted to assess the histological subtype, grade, depth of infiltration, resection margins (R0 = no cancer at resection margins; R1 = microscopic residual cancer; R2 = macroscopic residual cancer or M1), and nodal involvement. Treatment response was evaluated using the AJCC Tumor Regression Grade (TRG) system. This four-grade system classifies responses as follows: TRG 0 indicates a complete response with no viable cancer cells; TRG 1 represents a near-complete response with single cells or rare small groups of cancer cells; TRG 2 signifies a partial response with residual cancer showing evident tumor regression; and TRG 3 denotes a poor or no response, characterized by extensive residual cancer without evident tumor regression.

### 2.5. Adjuvant Therapy

Post-surgical adjuvant chemotherapy and immunotherapy were administered based on the collaborative decision of oncologists and patients. The specific regimens and schedules were tailored according to individual patient factors and responses to neoadjuvant treatment.

### 2.6. Toxicity Assessment

Acute toxicities from neoadjuvant concurrent chemoradiotherapy were evaluated weekly by physicians using CTCAE v4.0. The severity and frequency of adverse events were documented and comparatively analyzed between treatment groups.

### 2.7. Follow-Up and Survival Assessment

Patient follow-up was conducted every 3 months for the first three years post-treatment, followed by biannual assessments. Each assessment included endoscopic examination, PET/CT, and/or chest CT. Patients were evaluated for late toxicities and disease recurrence. All mortality events were documented throughout the follow-up period to ensure accurate survival data collection.

### 2.8. Statistical Analyses

The primary endpoints were pathologic complete response (pCR), progression-free survival (PFS), and overall survival (OS). PFS was defined as the interval from biopsy to disease recurrence or last follow-up, while OS was measured from biopsy to all-cause death or last follow-up. Survival outcomes were estimated using the Kaplan–Meier method, with inter-group differences assessed by the log-rank test. Univariate Cox proportional hazards models were employed to estimate hazard ratios and 95% confidence intervals for potential prognostic factors, including sex, age, margin status, angiolymphatic invasion (ALI), perineural invasion (PNI), and grade (well-to-moderately differentiated vs. poorly differentiated).

Categorical variables were compared between groups using the chi-square test or Fisher’s exact test, as appropriate. The Mann–Whitney U test was used to compare median ages between groups. Statistical significance was set at *p* < 0.05. All analyses were performed using SPSS software version 25 (IBM Corp., Armonk, NY, USA). ChatGPT-4 by OpenAI (May 2025 version) was used to assist with manuscript writing. 

## 3. Results

A total of 91 patients were included in this retrospective cohort study, with 61 patients in the standard chemotherapy group and 30 patients in the weekly chemotherapy group. Table 1 summarizes the patients’ characteristics, which were generally well-balanced between the groups.

Both groups were predominantly male (57 patients, 93.4%, in standard group vs. 28 patients, 93.3%, in weekly group; *p* = 1.000). The median age was 61.0 years (IQR: 54.0–65.5) in the standard group and 59.0 years (IQR: 52.8–63.0) in the weekly group (*p* = 0.560).

Squamous cell carcinoma was the predominant histological type in both groups, accounting for 59 patients (96.7%) in the standard group and 27 patients (90.0%) in the weekly group (*p* = 0.333). Most tumors were grade 2, with similar proportions between the standard group (58 patients, 95.1%) and weekly group (27 patients, 90.0%). Grade 3 tumors were slightly more common in the weekly group (three patients, 10.0%) compared to the standard group (two patients, 3.3%), while grade 1 tumors were rare, found in only one patient (1.6%) in the standard group and none in the weekly group.

Tumor location was classified based on the position of the tumor epicenter: upper (from the cervical esophagus to the lower border of the azygos vein), middle (from the lower border of the azygos vein to the lower border of the inferior pulmonary vein), and lower (from the lower border of the inferior pulmonary vein to the stomach, including the gastroesophageal junction). Lower esophageal tumors predominated in both groups, although more so in the weekly group (23 patients, 76.7%) than the standard group (39 patients, 63.9%). Middle esophageal tumors were more frequent in the standard group (22 patients, 36.1%) compared to the weekly group (6 patients, 20.0%). Upper esophageal tumors were rare, with none in the standard group and only one patient (3.3%) in the weekly group.

The clinical staging indicated that most patients had locally advanced disease at diagnosis. In the standard chemotherapy group, 54 patients (88.5%) were classified as clinical stage III, compared to 22 patients (73.3%) in the weekly chemotherapy group. Conversely, clinical stage IV disease was more frequently observed in the weekly chemotherapy group (seven patients, 23.3%, vs. four patients, 6.6%), although this difference did not reach statistical significance (*p* = 0.069). Distant metastasis (M1) was observed only in the weekly group (6.7% vs. 0%; *p* = 0.106). The overall distribution of clinical stages suggests a trend toward more advanced disease in the weekly chemotherapy group. This imbalance may be attributed to the impact of the COVID-19 pandemic, as delays in diagnosis and treatment initiation during this period likely contributed to the higher proportion of more advanced disease that was observed in the weekly group.

The pathologic staging after neoadjuvant therapy showed similar distributions between groups. Stage I disease was the most common in both the standard and weekly chemotherapy groups (29 patients, 47.5%, vs. 15 patients, 50.0%), followed by stage III (17 patients, 27.9%, vs. 8 patients, 26.7%), with stages II and IV being less frequent. A slightly higher proportion of pathologic stage IV was observed in the weekly group (four patients, 13.3%, vs. three patients, 4.9%). The rates of positive surgical margins were low (four patients, 6.6%, vs. one patient, 3.3%; *p* = 1.000). Angiolymphatic invasion (ALI) and perineural invasion (PNI) were more frequently seen in the weekly group (ALI: six patients, 20.0%, vs. seven patients, 11.5%; *p* = 0.342. PNI: five patients, 16.7%, vs. seven patients, 11.5%; *p* = 0.521), although this was not statistically significant.

The treatment response rates (CR + PR) were comparable between the groups (90.2% vs. 86.7%; *p* = 0.724). However, the complete response (CR) rate was higher in the weekly chemotherapy group (40.0% vs. 26.2%), although this difference was not statistically significant (*p* = 0.181).

Notably, the standard chemotherapy group experienced a significantly higher recurrence rate (52.5% vs. 26.7%; *p* = 0.020) and a longer median time to recurrence (20.1 months vs. 9.1 months; *p* < 0.001). The mortality rate was also higher in the standard chemotherapy group (34.4% vs. 6.7%; *p* = 0.004). However, these differences should be interpreted in the context of follow-up duration, as the median follow-up time was significantly longer in the standard chemotherapy group (22.8 months vs. 10.6 months; *p* < 0.001). This extended follow-up period in the standard chemotherapy group may account for the observed differences in recurrence rates, time to recurrence, and mortality.

Univariate and multivariate analyses were conducted to identify potential prognostic factors for progression-free survival (PFS) and overall survival (OS) (Table 2). In the multivariate Cox proportional hazards regression analysis, perineural invasion (coded as binary yes/no) was examined alongside the following covariates: gender, age, positive margin, angiolymphatic invasion, clinical staging, treatment response, and treatment group. For PFS, the univariate analysis identified perineural invasion (PNI) (HR 3.72, 95% CI 1.78–7.76; *p* < 0.001) and non-complete response (non-CR) (HR: 2.62; 95% CI: 1.16–5.90; *p* = 0.020) as being associated with worse PFS. In the multivariate analysis, PNI remained a significant independent predictor of worse PFS (HR: 2.83; 95% CI: 1.33–6.01; *p* = 0.007), while non-CR showed a trend towards significance (HR: 2.14; 95% CI: 0.93–4.95; *p* = 0.075).

For OS, the univariate analysis revealed PNI as the only significant prognostic factor (HR: 4.75; 95% CI: 1.76–12.83; *p* = 0.002). However, no factors remained significant in the multivariate analysis for OS. Notably, the choice of chemotherapy regimen (standard chemotherapy group vs. weekly chemotherapy group) did not significantly impact PFS (HR: 1.11; 95% CI: 0.52–2.39; *p* = 0.787) or OS (HR: 0.60; 95% CI: 0.13–2.71; *p* = 0.507) in the univariate analysis.

Kaplan–Meier analyses were performed to assess OS and PFS for patients in the standard chemotherapy group and weekly chemotherapy group (Figure 2 and Figure 3). The 18-month PFS rates were 54.1% for the standard group (median PFS 25.9 months) and 46.5% for the weekly group (median PFS 16.4 months). The 18-month OS rates were 77.0% and 89.8% for the standard and weekly groups, respectively. Despite these descriptive differences, log-rank tests revealed no statistically significant differences between the groups for either OS (*p* = 0.503) or PFS (*p* = 0.892). The slight difference in 18-month PFS rates between groups is likely due to random variation and does not reflect a statistically meaningful disparity given the overlapping survival curves and limited sample size.

While survival outcomes are crucial, treatment tolerability and adherence are equally important factors in assessing the efficacy of CCRT regimens. In our study, we observed a favorable toxicity profile and high compliance rates across both treatment groups. The most common acute toxicities observed were dysphagia and hematology-related adverse events (Table 3). Grade 1–2 dysphagia occurred in 24.59% of patients in the standard chemotherapy group and 26.67% in the weekly chemotherapy group. Hematologic toxicities, primarily leukopenia and anemia, were also prevalent, with Grade 1–2 leukopenia affecting 8.20% and 16.67% of patients in the standard and weekly groups, respectively. Grade 1–2 anemia affected 11.48% and 16.67% in the respective groups. Grade 3–4 toxicities were infrequent, with leukopenia being the most common (3.28% in the standard group and 0% in the weekly group). Other grade 3–4 toxicities were rare, with only one case of grade 3 liver enzyme elevation in the weekly chemotherapy group. Overall, acute toxicities were comparable between the two groups; however, when adverse events occurred, those in the standard group tended to be of slightly higher severity, particularly for hematologic toxicities, with a higher incidence of grade 2–3 leukopenia. Regarding CCRT compliance, all 61 patients in the standard group completed the treatment as defined, yielding a 100% compliance rate. In the weekly group, 29 out of 30 patients (96.67%) met the completion criteria, with only one patient discontinuing after two cycles due to personal preference. This patient experienced only mild toxicities (grade 1 dysphagia and grade 2 fatigue), along with moderate declines in hematologic parameters (WBC dropped from 6330 to 3440; Hb from 14.2 to 12.3), without any severe complications. These high compliance rates in both groups suggest that the treatment regimens were generally well-tolerated.

Considering that no statistically significant differences were observed in survival outcomes, toxicity profiles, or compliance rates between the two treatment arms, these findings suggest that concurrent weekly chemotherapy may be a feasible alternative to the standard chemotherapy regimen, based on the comparable clinical outcomes.

## 4. Discussion

Our investigation of weekly versus standard neoadjuvant chemoradiotherapy for locally advanced esophageal cancer was motivated by successful applications in head and neck squamous cell carcinoma (HNSCC). HNSCC and esophageal squamous cell carcinoma share histological similarities due to squamous differentiation. The COVID-19 pandemic underscored the need for treatment strategies that minimize hospital exposure, making outpatient weekly chemotherapy particularly attractive. No direct comparison of weekly versus standard schedules in esophageal cancer patients has previously been conducted, representing a significant knowledge gap, which our study addresses.

The synergistic effects of concurrent chemotherapy and radiation stem from complementary mechanisms targeting different phases of the cell cycle. Radiation sensitivity varies by phase, being highest in G2/M and lowest in the S phase, where DNA repair is most active. Cisplatin forms DNA crosslinks primarily during the S phase, sensitizing cells to radiation damage, while 5-fluorouracil inhibits thymidylate synthase and induces S-phase arrest, synchronizing surviving cells into the more radiosensitive G2/M phase. Traditionally, 5-FU continuous infusion over four days was thought to maintain steady therapeutic levels for optimal radiosensitization across multiple radiation fractions, given 5-FU’s short half-life of approximately 10 min after bolus injection. However, this theoretical advantage has not been consistently supported by clinical outcomes. The current evidence suggests that achieving minimum effective cumulative doses is more critical than maintaining continuous drug exposure, with weekly dosing potentially offering superior timing that matches radiation fractionation schedules that are delivered over 5–6 weeks. Furthermore, intermittent dosing may prevent the selection of resistant tumor clones that could emerge under continuous high-dose exposure. This coordinated targeting across multiple checkpoints enhances cytotoxicity compared to either modality alone.

Weekly chemotherapy schedules offer several practical advantages over standard regimens, including reduced acute toxicity, improved patient compliance, and decreased hospitalization requirements. The approach allows for more frequent monitoring of treatment tolerance and earlier detection of adverse effects. The current evidence suggests that achieving minimum effective cumulative doses is more critical than the cycle intensity, with high-dose regimens potentially increasing toxicity without clear survival benefits over weekly approaches. In our study, both regimens demonstrated excellent compliance rates (100% vs. 96.67%), with comparable overall toxicity profiles, although the standard group experienced slightly higher severity grade 2–3 hematologic toxicities. The single discontinuation in the weekly group was due to personal preference rather than severe toxicity, indicating that both treatments were well-tolerated. This suggests that the observed superiority in outcomes is not primarily attributable to differential treatment completion or toxicity-related dose reductions, pointing to alternative biological mechanisms underlying the weekly regimen’s efficacy.

Previous studies in esophageal cancer support the feasibility of the weekly regimen [9,10]. Pasini et al. demonstrated a 47% pathological complete response rate using weekly docetaxel and cisplatin with continuous 5-FU, while Chen et al. reported a 68.8% objective response rate with weekly cisplatin and 5-FU in postoperative recurrence, showing improved toxicity profiles [11,12]. Our study extends these findings to the neoadjuvant setting with modified dosing.

Evidence from HNSCC consistently demonstrates weekly cisplatin’s efficacy to be comparable to standard 3-weekly schedules [13,14]. Studies by Sahoo et al., Rawat et al., and Ghosh et al. showed similar response rates and locoregional control with weekly regimens, while reducing toxicity and improving compliance [15,16,17]. Espeli et al. found weekly cisplatin to be less nephrotoxic with similar progression-free survival rates, and Veldman et al. reported no difference in functional outcomes between dosing schedules [18,19]. These findings support weekly cisplatin as non-inferior to standard regimens in terms of oncological outcomes, while offering reduced acute toxicity.

Our study of 91 esophageal squamous cell carcinoma (SCC) patients demonstrated comparable treatment response rates between the weekly and standard groups (90.2% vs. 86.7%; *p* = 0.724), suggesting maintained efficacy with reduced hospital exposure. Notably, the weekly group had a higher proportion of advanced-stage disease at baseline (though not statistically significant), yet still achieved comparable response rates and favorable early outcomes, further supporting the potential utility of this regimen. The comparable compliance rates between groups (100% vs. 96.67%) indicate that the superior outcomes in the weekly group cannot be attributed to better treatment completion, supporting the hypothesis that cumulative dose delivery over the entire radiation course may be more important than maintaining continuous drug exposure during individual cycles. While the higher complete response rate in the weekly group (40.0% vs. 26.2%) and significantly lower recurrence (26.7% vs. 52.5%; *p* = 0.020) and mortality rates (6.7% vs. 34.4%; *p* = 0.004) are promising, these findings require cautious interpretation given differences in follow-up duration and the retrospective design.

This study has important limitations, including its retrospective, single-center design, relatively small sample size, and unequal follow-up durations between groups. Patient allocation was determined by institutional policy timing rather than randomization, potentially introducing confounding factors. These findings should be interpreted as preliminary and descriptive rather than definitive. Future research should include larger, multi-center prospective studies with balanced follow-up periods to confirm our results and determine optimal weekly dosing regimens for esophageal cancer patients.

## 5. Conclusions

Our retrospective observational study shows that a weekly cisplatin and 5-FU regimen may provide comparable pathologic responses and favorable early outcomes to the standard regimen in locally advanced esophageal squamous cell carcinoma. This suggests that weekly chemotherapy could be a viable alternative during pandemics, maintaining treatment efficacy while reducing infectious risks. Further research is needed to confirm long-term outcomes, but this approach offers a promising strategy for patient safety in challenging healthcare environments.

## Figures and Tables

**Figure 1 medicina-61-01326-f001:**
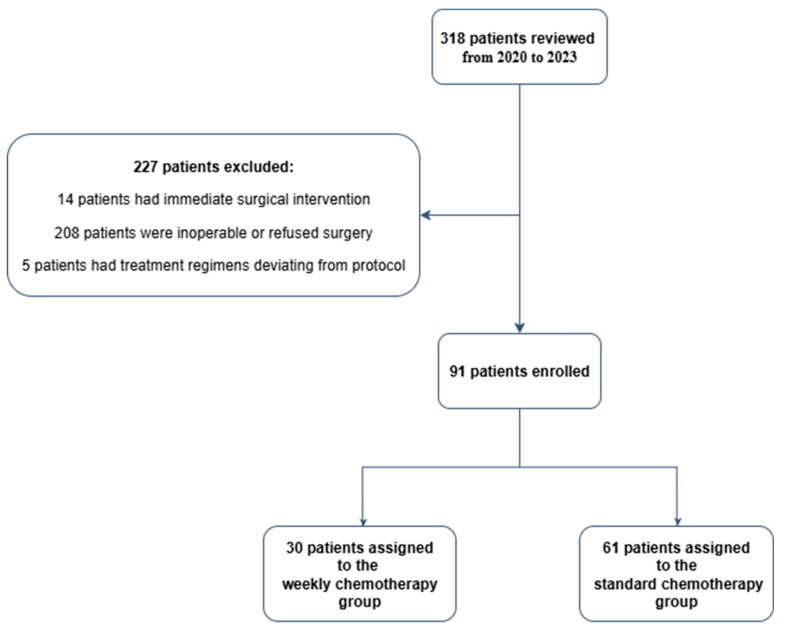
Study flow chart.

**Figure 2 medicina-61-01326-f002:**
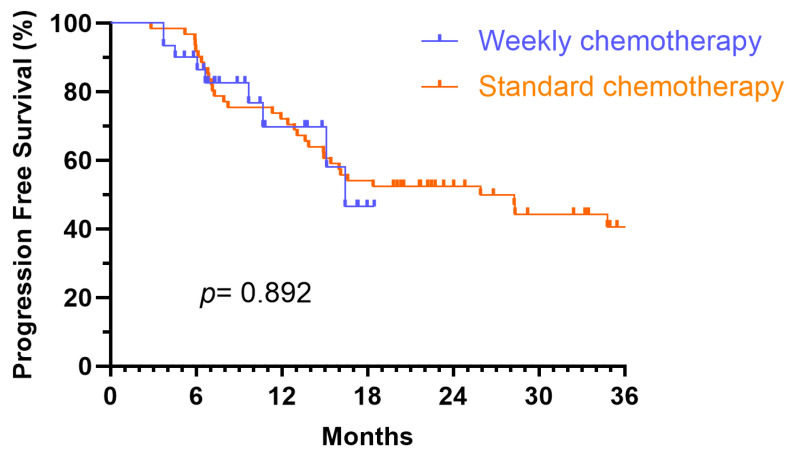
Kaplan–Meier survival curves for progression-free survival (PFS) among different chemotherapy schedules.

**Figure 3 medicina-61-01326-f003:**
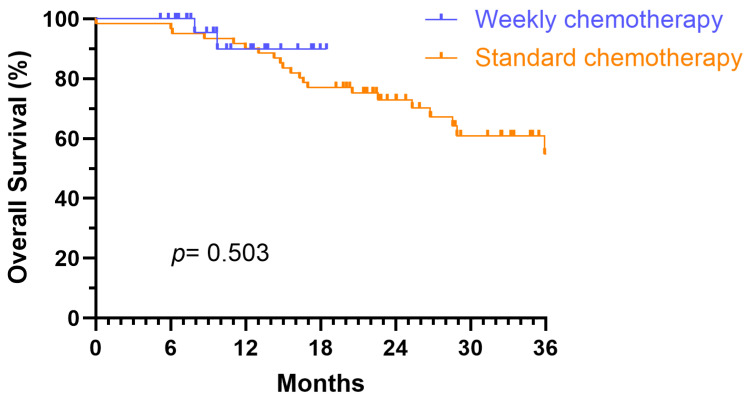
Kaplan–Meier survival curves for overall survival (OS) among different chemotherapy schedules.

**Table 1 medicina-61-01326-t001:** Patient characteristics (n = 91).

	Standard Chemotherapy (n = 61)	Weekly Chemotherapy (n = 30)	*p* Value
Gender			1.000
Female	4 (6.6%)	2 (6.7%)	
Male	57 (93.4%)	28 (93.3%)	
Age	61.0 (54.0–65.5)	59.0 (52.8–63.0)	0.560
Histology			0.333
Squamous cell carcinoma	59 (96.7%)	27 (90.0%)	
Adenocarcinoma	2 (3.3%)	3 (10.0%)	
Grade			0.333
G1	1 (1.6%)	0 (0%)	
G2	58 (95.1%)	27 (90%)	
G3	2 (3.3%)	3 (10%)	
Tumor location			0.123
Upper	0 (0%)	1 (3.3%)	
Middle	22 (36.1%)	6 (20.0%)	
Lower	39 (63.9%)	23 (76.7%)	
Post-neoadjuvant therapy staging			0.443
I	29 (47.5%)	15 (50.0%)	
II	12 (19.7%)	3 (10.0%)	
III	17 (27.9%)	8 (26.7%)	
IV	3 (4.9%)	4 (13.3%)	
Clinical staging			0.069
I	0 (0%)	0 (0%)	
II	3 (4.9%)	1 (3.3%)	
III	54 (88.5%)	22 (73.3%)	
IV	4 (6.6%)	7 (23.3%)	
Positive margin	4 (6.6%)	1 (3.3%)	1.000
ALI	7 (11.5%)	6 (20.0%)	0.342
PNI	7 (11.5%)	5 (16.7%)	0.521
Treatment response			0.724
CR + PR	55 (90.2%)	26 (86.7%)	
SD + PD	6 (9.8%)	4 (13.3%)	
Treatment response (classified as complete or incomplete)			0.181
CR	16 (26.2%)	12 (40.0%)	
Non-CR	45 (73.8%)	18 (60.0%)	
Recurrence	32 (52.5%)	8 (26.7%)	0.020 *
Relapse interval	20.1 (9.8–30.8)	9.1 (6.6–14.0)	<0.001 **
Death	21 (34.4%)	2 (6.7%)	0.004 **
Follow-up duration	22.8 (19.5–32.9)	10.6 (7.5–14.0)	<0.001 **

PNI: perineural invasion; ALI: angiolymphatic invasion; CR: complete response; PR: partial response; SD: stable disease; PD: progressive disease. Chi-square test/Fisher’s exact test or Mann–Whitney U test. Median (IQR) * *p* < 0.05, ** *p* < 0.01.

**Table 2 medicina-61-01326-t002:** Univariate and multivariate analyses of prognostic factors for progression-free survival (PFS).

	Univariate			Multivariable		
	HR	95% CI	*p* Value	HR	95% CI	*p* Value
Gender						
Female	Reference					
Male	4.06	(0.56–29.57)	0.167			
Age	1.01	(0.97–1.04)	0.747			
Positive margin	2.68	(0.94–7.64)	0.064			
ALI	1.98	(0.91–4.32)	0.085			
PNI	3.72	(1.78–7.76)	<0.001 **	2.83	(1.33–6.01)	0.007 **
Clinical staging						
II	Reference					
III	1.00	(0.24–4.18)	0.996			
IV	0.88	(0.16–4.82)	0.878			
Treatment response						
CR + PR	Reference					
SD + PD	2.26	(0.99–5.14)	0.053			
Treatment response (classified as complete or incomplete)						
CR	Reference			Reference		
Non-CR	2.62	(1.16–5.90)	0.020 *	2.14	(0.93–4.95)	0.075
Treatment group						
Weekly chemotherapy group	Reference					
Standard chemotherapy group	1.11	(0.52–2.39)	0.787			

Cox proportional hazard regression. * *p* < 0.05, ** *p* < 0.01. PNI: perineural invasion; ALI: angiolymphatic invasion; CR: complete response; PR: partial response; SD: stable disease; PD: progressive disease.

**Table 3 medicina-61-01326-t003:** Toxicity and compliance.

	Standard Chemotherapy (n = 61)	Weekly Chemotherapy (n = 30)
Leukopenia		
Grade 1	3 (4.92%)	5 (16.67%)
Grade 2	2 (3.28%)	0 (0%)
Grade 3	2 (3.28%)	0 (0%)
Grade 4	0 (0%)	0 (0%)
Grade 5	0 (0%)	0 (0%)
Anemia		
Grade 1	6 (9.84%)	5 (16.67%)
Grade 2	1 (1.64%)	0 (0%)
Grade 3	0 (0%)	0 (0%)
Grade 4	0 (0%)	0 (0%)
Grade 5	0 (0%)	0 (0%)
Thrombocytopenia		
Grade 1	5 (8.20%)	4 (13.33%)
Grade 2	0 (0%)	0 (0%)
Grade 3	0 (0%)	0 (0%)
Grade 4	0 (0%)	0 (0%)
Grade 5	0 (0%)	0 (0%)
Dysphagia		
Grade 1	11 (18.03%)	6 (20%)
Grade 2	4 (6.56%)	2 (6.67%)
Grade 3	0 (0%)	0 (0%)
Grade 4	0 (0%)	0 (0%)
Grade 5	0 (0%)	0 (0%)
Cough		
Grade 1	1 (1.64%)	0 (0%)
Grade 2	2 (3.28%)	0 (0%)
Grade 3	0 (0%)	0 (0%)
Grade 4	0 (0%)	0 (0%)
Grade 5	0 (0%)	0 (0%)
Liver enzyme elevation		
Grade 1	1 (1.64%)	1 (3.33%)
Grade 2	0 (0%)	0 (0%)
Grade 3	0 (0%)	1 (3.33%)
Grade 4	0 (0%)	0 (0%)
Grade 5	0 (0%)	0 (0%)
Creatinine elevation		
Grade 1	1 (1.64%)	1 (3.33%)
Grade 2	0 (0%)	0 (0%)
Grade 3	0 (0%)	0 (0%)
Grade 4	0 (0%)	0 (0%)
Grade 5	0 (0%)	0 (0%)
Nausea/vomiting		
Grade 1	4 (6.56%)	2 (6.67%)
Grade 2	0 (0%)	1 (3.33%)
Grade 3	0 (0%)	0 (0%)
Grade 4	0 (0%)	0 (0%)
Grade 5	0 (0%)	0 (0%)
Diarrhea		
Grade 1	0 (0%)	1 (3.33%)
Grade 2	0 (0%)	0 (0%)
Grade 3	0 (0%)	0 (0%)
Grade 4	0 (0%)	0 (0%)
Grade 5	0 (0%)	0 (0%)
Dermatitis		
Grade 1	2 (3.28%)	0 (0%)
Grade 2	1 (1.64%)	0 (0%)
Grade 3	0 (0%)	0 (0%)
Grade 4	0 (0%)	0 (0%)
Grade 5	0 (0%)	0 (0%)
Radiotherapy compliance		
<5% dose deviation	61 (100%)	30 (100%)
≥5% dose deviation	0 (0%)	0 (0%)
Cumulative chemotherapy dose (mg)		
Cisplatin (average cumulative dose)	254.03	160.17
5FU (average cumulative dose)	10,591.48	5976.93
CCRT compliance		
Complete CCRT course	61 (100%)	29 (96.67%)
Incomplete CCRT course	0 (0%)	1 (3.33%)

## Data Availability

The data presented in this study are available on request from the corresponding author. The data are not publicly available due to privacy or ethical restrictions.

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
