# Peer review of "Weekly Cisplatin and 5-Fluorouracil in Neoadjuvant Chemoradiotherapy for Esophageal Cancer: A Pandemic-Era Evaluation"

_medicina, 2025, doi:10.3390/medicina61081326_

Round 1
Reviewer 1 Report
Comments and Suggestions for Authors
I would like to thank the authors for their work. The introduction and discussion sections are well written, but my concerns about the material-method and conclusion sections are as follows.
How did you determine the treatment completion criteria?
Wasn't the most commonly used CROSS regimen in the literature abandoned during this period? If it was used, I suggest you compare it with the CROSS regimen.
Also, if weekly treatment was intended, I suggest explaining why weekly platinum-5FU was preferred instead of the CROSS regimen.
Disease localizations (middle or lower) were not given, I suggest you share this data.
Although not statistically significant (since the number of possible patients is small), there are significant differences in terms of disease characteristics (most importantly stage) between the 2 groups of patients.
The median follow-up period is significantly different between the arms, it would not be correct to compare recurrence rates with this data.
‘’ Our findings demonstrate that the weekly chemotherapy schedule is non-inferior to the standard regimen re- 240 guarding survival outcomes for patients with locally advanced esophageal ‘’ to establish this sentence, a randomized non-inferiority study is necessary.
Best regards
Reviewer 2 Report
Comments and Suggestions for Authors
The authors compared weekly versus standard monthly cisplatin and 5-FU regimens for neoadjuvant chemoradiotherapy in esophageal squamous cell carcinoma during the COVID-19 pandemic, finding that the weekly regimen produced comparable efficacy while exhibiting lower recurrence and mortality rates, indicating it may serve as a viable alternative during pandemic .
The methods are well described, and the results and conclusions are presented clearly.
A few things that need attention are:
- Line 33 – Please introduce the full form of the abbreviation CROSS. It is ChemoRadiotherapy for Oesophageal cancer Followed by Surgery
- Line 75 - Please introduce the full form of the abbreviation FDG/PET
- It is important for the authors to comment on the KM plots explaining the lower PFS associated with weekly chemotherapy, even though it is not significant.
Reviewer 3 Report
Comments and Suggestions for Authors
- The manuscript is not formatted according to requirements.
- There is no reference at the end of the COVID chapter in Introduction. Please also include literature data how the pandemic affected these patients' treatment process.
- Pathologically confirmed - Please rephrase to histologically confirmed.
- Histology type - Please rephrase to histological subtype.
- Tumor extension - Does that mean tumor size, aka largest tumor diameter? Or depth of infiltration?
- Resection margins - Meaning are they free of tumor?
- How come that not AJCC and Mandard regression scores were used?
- The Ryan tumor regression grade system should not be detailed like this.
- Table 1 showed the patients’ characteristics. - Rephrase to summarizes. And patient characteristics should be detailed in the text, as well.
- Tumor localisation should be included, as well, while it is determined by the histological subtype, as well, and it largely influences the prognosis.
- What was the tumor grade?
- Table 1 contains SCC as an abbreviation, while it not yet defined.
- Why is there Clinical T, N, M, and then Clinical TNM in Table 1?
- None of the abbreviations used in Table 1 are defined.
- What is ALI/PNI?
- Results - When you describe numbers, please do not just include percentages. The exact numbers should be included, as well.
- Table 2 - Arm?
- Why are there different lettering types used?
- In multivariate analysis, what was perineural invasion examined with?
- Style of Table 3 is completely different.
- How were the adverse effects graded? How come they all divided into 4 categories?
- Considering the comparable survival outcomes, manageable toxicity profiles, and high compliance rates observed in both treatment arms, these findings suggest that concurrent weekly chemotherapy may be a viable alternative to the standard chemotherapy regimen. - There were no significant differences found in between these two, so how come the authors came to this conclusion?
- ... head and neck squamous cell carcinoma (HNSCC), which shares histological similarities with esophageal squamous cell carcinoma - Which ones exactly?
- There are no exact resuls included in the discussion section.
- There are not enough references.
- And the references are not formatted according to the requirements.
Reviewer 4 Report
Comments and Suggestions for Authors
This single-center retrospective study compares a weekly versus a standard monthly regimen of cisplatin and 5-fluorouracil in neoadjuvant chemoradiotherapy for patients with locally advanced esophageal squamous cell carcinoma. Initiated during the COVID-19 pandemic to reduce inpatient burden, the weekly regimen appeared to yield higher rates of pathologic complete response, lower recurrence, and significantly reduced mortality. The study suggests that weekly chemotherapy may offer a viable alternative for treatment delivery under resource-constrained or outpatient-prioritized setting.
This is a relevant and timely manuscript evaluating a pragmatic modification to neoadjuvant chemoradiotherapy regimens for esophageal cancer under pandemic-related constraints. The authors present comparative data on weekly versus standard chemotherapy protocols, with attention to outcomes including pCR, recurrence, and overall survival. The findings are intriguing—particularly the lower recurrence and mortality seen in the weekly group—but several key issues limit the interpretability and strength of the conclusions.
First, it must be made more explicit that this is a retrospective observational study, not a randomized controlled trial. Although the “group assignment” followed hospital policy change, this temporal allocation introduces potential confounding by indication, time bias, and differences in follow-up duration, all of which are only superficially addressed. The shorter follow-up in the weekly cohort (10.6 vs. 22.8 months) is particularly important in interpreting survival and recurrence data—especially since the mortality benefit is striking but may be skewed by limited observation time.
Second, the apparent superiority of weekly chemotherapy, especially in mortality and recurrence, is not adequately explored mechanistically. Could the reduced dose intensity or altered pharmacokinetics of weekly cisplatin/5-FU better synergize with radiotherapy? Was toxicity reduced in a way that allowed better treatment completion? These aspects deserve more discussion, as they underpin any biologic rationale for the observed difference.
Additionally, the manuscript would benefit from:
-
Clarifying patient selection criteria beyond institutional timing (e.g., baseline stage, nutritional status, comorbidities).
-
Explicitly stating in both abstract and methods that this was a retrospective observational study, and highlighting this limitation more forcefully in the discussion.
-
A Kaplan-Meier curve for PFS, not just OS, given its relevance for recurrence.
-
An adjusted analysis or at least a table comparing baseline characteristics between groups would help assess selection bias.
-
Was this a randomized controlled trial? No—it was an observational, retrospective study. This must be clearly and consistently stated throughout the manuscript, especially in the abstract and methods.
-
The weekly regimen appears superior in recurrence and mortality, yet this is not adequately dissected. Is this truly a treatment effect, or the result of immortal time bias or shorter follow-up in the weekly cohort?
-
Authors should explore toxicity, treatment adherence, and possible radiobiological synergy hypotheses to support these findings beyond the descriptive statistics.
Round 2
Reviewer 1 Report
Comments and Suggestions for Authors
I would like to thank the authors who responded to my numerous criticisms to my satisfaction and added the corrections to their articles. I believe that it will contribute to the literature in its current form. Best regards
Author Response
Response to Reviewer 1
Comments : I would like to thank the authors who responded to my numerous criticisms to my satisfaction and added the corrections to their articles. I believe that it will contribute to the literature in its current form.
Response : We wound like to thank the reviewer for your kind words and for taking the time to provide such thoughtful feedback throughout the review process. Your comments greatly help us improved the clarity and quality of our manuscript, and we truly appreciated your detailed suggestions. We are grateful for your support in moving it forward.Please don’t hesitate to share any further recommendations you may have in the future—we would welcome your insights.
Reviewer 3 Report
Comments and Suggestions for Authors
Thank you for carrying out the recommended modifications!
A few more minor details I found are as follows:
- For instance, one study - Please rephrase.
- Both HNSCC and esophageal squamous cell carcinoma share histological similarities as squamous cell carcinomas arising from stratified squamous epithelium. Please rephrase to: HNSCC and esophageal squamous cell carcinoma share histological similarities due to squamous differentiation.
- In Materials and methods, please change header Patients to Patient selection.
- Figure 1 needs some adjustments while some of the components include a number upfront, while some include numbers as n=... - Please unify.
- Please change Pathological analyses to Histopathological evaluation. Grade instead of histologic grade.
- Histological differentiation in the Statistical Analyses section - that is grade!
- classified by complete response or not - Please rephrase to complete or incomplete
- Follow up time - Does this mean overall survival? Because later on OS and PFS are mentioned.
- How was OS and PFS defined?
Author Response
Response to Reviewer 3
Comments 1: For instance, one study - Please rephrase.
Response 1: Thank you for the suggestion. The sentence has been revised to: "One study from the United Kingdom showed that 56.5% of centers omitted chemotherapy from concurrent chemoradiotherapy for head and neck cancer patients to limit exposure risks[8]."
Comments 2: Both HNSCC and esophageal squamous cell carcinoma share histological similarities as squamous cell carcinomas arising from stratified squamous epithelium. Please rephrase to: HNSCC and esophageal squamous cell carcinoma share histological similarities due to squamous differentiation
Response 2: Thank you for the suggestion. The sentence has been revised as recommended to: "HNSCC and esophageal squamous cell carcinoma share histological similarities due to squamous differentiation."
Comments 3: In Materials and methods, please change header Patients to Patient selection.
Response 3: Thank you for the suggestion. The header "Patients" has been changed to "Patient Selection" in the Materials and Methods section.
Comments 4: Figure 1 needs some adjustments while some of the components include a number upfront, while some include numbers as n=... - Please unify.
Response 4: Thank you for pointing this out. Figure 1 has been revised to unify the formatting, with all components now consistently displaying numbers in the same format throughout the flowchart.
Comments 5: Please change Pathological analyses to Histopathological evaluation. Grade instead of histologic grade.
Response 5:Thank you for the suggestion. "Pathological analyses" has been changed to "Histopathological evaluation" and "histologic grade" has been changed to "Grade" throughout the manuscript.
Comments 6: Histological differentiation in the Statistical Analyses section - that is grade!
Response 6: Thank you for the clarification. "Histological differentiation" has been changed to "grade" in the Statistical Analyses section.
Comments 7: classified by complete response or not - Please rephrase to complete or incomplete
Response 7:Thank you for the suggestion. The phrase "classified by complete response or not" has been rephrased to "classified as complete or incomplete”.
Comments 8: Follow up time - Does this mean overall survival? Because later on OS and PFS are mentioned.
Response 8:Thank you for the question. To clarify, "follow-up time" refers to the follow-up duration of the study, which is different from survival outcomes. The OS and PFS mentioned later are the survival endpoints measured during this follow-up period. The table has been revised to change "follow-up time" to "follow-up duration" for clarity.
Comments 9: How was OS and PFS defined?
Response 9: Thank you for the question. The definitions of OS and PFS are provided in the manuscript at lines 158-159. PFS was defined as the interval from biopsy to disease recurrence or last follow-up, while OS was measured from biopsy to all-cause death or last follow-up. If you would like these definitions emphasized elsewhere in the manuscript or require any additional clarification, please let us know and we will be happy to make the necessary revisions. Thank you again for your time, effort, and valuable recommendations.